# GENAUDIT: Fixing Factual Errors in Language Model Outputs with Evidence

**Kundan Krishna**♣     **Sanjana Ramprasad**◇     **Prakhar Gupta**♣
**Byron C. Wallace**◇     **Zachary C. Lipton**♣     **Jeffrey P. Bigham**♣
♣ Carnegie Mellon University
◇ Northeastern University
{kundank,prakharg,zlipton,jbigham}@andrew.cmu.edu
{ramprasad.sa,b.wallace}@northeastern.edu

## Abstract

LLMs can generate factually incorrect statements even when provided access to reference documents. Such errors can be dangerous in high-stakes applications (e.g., document-grounded QA for healthcare or finance). We present GENAUDIT — a tool intended to assist fact-checking LLM responses for document-grounded tasks. GENAUDIT suggests edits to the LLM response by revising or removing claims that are not supported by the reference document, and also presents evidence from the reference for facts that do appear to have support. We train models to execute these tasks, and design an interactive interface to present suggested edits and evidence to users. Comprehensive evaluation by human raters shows that GENAUDIT can detect errors in 8 different LLM outputs when summarizing documents from diverse domains. To ensure that most errors are flagged by the system, we propose a method that can increase the error recall while minimizing impact on precision. We release our tool (GENAUDIT) and fact-checking model for public use.[1]

## 1 Introduction

LLMs can produce factually incorrect or unsubstantiated statements (Li et al., 2023; Min et al., 2023), even when they are explicitly provided relevant context such as documents (Adams et al., 2023; Sadat et al., 2023). Incidentally, such *document-grounded* generation is often involved in high-stakes usage scenarios where factual correctness is (especially) paramount. For example, a doctor using an LLM to summarize a patient's medical history (Adams et al., 2023; Kanwal and Rizzo, 2022) might make an incorrect decision if the generated summary contains errors. Manually verifying LLM outputs in such settings is therefore prudent, but also time-consuming and so undercuts the motivation for using language technologies in the first place. This motivates the need for a system that can *assist* users in efficiently verifying LLM output.

To this end, we introduce GENAUDIT, a tool for fact-checking LLM responses in document-grounded tasks such as summarization and question answering. Given a document and an LLM-generated output conditioned on the same, GENAUDIT (i) locates factual errors in the output text and proposes edits to fix them, and (ii) displays evidence to support facts in the (potentially edited) text. The system consists of two components: an interactive interface which presents evidence and edit suggestions for the user to act upon, and a bespoke backend model (fine-tuned LLM) capable of producing edits and identifying evidence. The interface allows the user to make edits to the LLM-generated text, and then observe updated predictions from the fact-checking model. Notably, in addition to supporting the

---

[1] https://github.com/kukrishna/genaudit

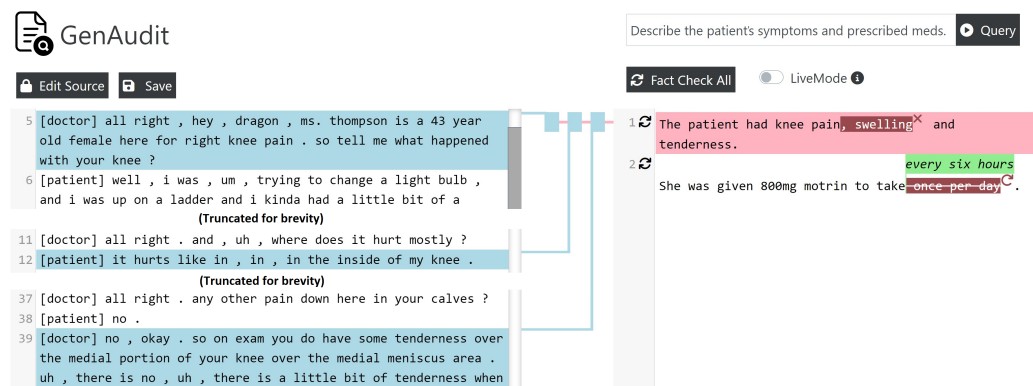

Figure 1: An illustration of GENAUDIT's user interface and sample predictions. Reference document (a clinical transcript) is on the left and the generated text to be fact-checked is on the right (generated by querying any LLM, but manually entered here for ease of illustration). Spans in the text which are not supported or are contradicted by the reference are highlighted in red, with suggested replacements in green. As the user moves to any line in the generated text, evidence found for all facts in it are highlighted using blue links. Evidence and error predictions shown here are made by a fine-tuned Flan-UL2 model backend.

task of fact-checking itself, the interface can also be used as a tool to evaluate and compare different backend fact-checking models, collecting data on human edits to fine-tune better models, and carry out counterfactual testing of fact-checking models by editing source documents.

We designed and evaluated different models to generate the fact-checking predictions for the tool, including fine-tuned and few-shot prompted LLMs. We treat this as a sequence-to-sequence task: Given an input document and a claim sentence, the model is required to simultaneously generate the sentence ids in the document which provide evidence, and a *revised* version of the claim which fixes any factual errors. We used data from the USB benchmark (Krishna et al., 2023) to train and evaluate models on the fact-checking tasks. We found that fine-tuned open-source LLMs perform better than few-shot prompted ChatGPT and GPT-4 models, at least when evaluated on an in-domain held-out test set.

Ideally, a fact-checking tool would support verifying text produced by any LLM, based on reference documents from any domain. We evaluated GENAUDIT using 8 different models to summarize documents from 3 different domains. Human annotators were asked to accept or reject edits suggested by the tool, fix errors that were not caught by it, and also to provide feedback on the usefulness of suggested evidence. On average, GENAUDIT highlighted ∼40% of erroneous words in summaries with a precision of ∼95%.[2] In terms of extracting useful evidence, GENAUDIT achieved ∼91% recall and ∼95% precision.

Human evaluations also show that GENAUDIT can be used to verify summarization outputs in different domains, including clinical conversations, news articles and social media posts. This is despite the fact-checking model being trained only on Wikipedia data.

GENAUDIT successfully identified errors in outputs from 8 different LLMs including Mistral-7B (Jiang et al., 2023), LLama2-70B (Touvron et al., 2023), Gemini-pro (Team et al., 2023) and GPT-4 (Achiam et al., 2023) in human evaluation. Observed precision ranged between 79 − 100% while, recall ranged from 23 − 57% for different generation models. Our human evaluation yielded a collection of 702 summaries generated by state-of-the-art models carefully annotated with factual errors; this may be useful for future research on fact-checking LLMs.

The relative trade-off between identifying errors (recall) and making efficient use of expert time (precision) will depend on the particular use-case. We therefore introduce a decoding algorithm for fact-checking models which generate revised/fixed versions of claims, which can increase the recall of error detection with minimal cost in precision. This approach entails intervening at time-steps

---

[2] For reference, ∼4% of words in summaries are erroneous, on average.

| |
|---|
| **Input:** |
| You are provided a document and its summary. The summary may potentially contain factual errors. The last sentence of the summary is marked as a claim. Find all sentences in the document providing evidence for the claim, and then revise the claim to remove or replace unsupported facts. DOCUMENT: SENT0 Micheal Ward SENT1 Early life. SENT2 Micheal Ward was born in Spanish Town, Jamaica on 18 November 1997. SENT3 His mother was 18 years old when he was born. SENT4 He has three sisters. ... SENT17 Ward's breakout year came in 2019, when he starred as Jamie in Netflix's revival and third series of "Top Boy". SENT18 He also appeared in a leading role in the film "Blue Story" in the same year. SENT19 The film received critical acclaim, and Ward won the BAFTA Rising Star Award for his performance. ... SUMMARY: Micheal Ward (born 18 November 1997) is a Jamaican-British actor and former model. CLAIM: His films include "Blue Story" (2018) and "The Old Guard" (2020). |
| **Output:** |
| EVIDENCE: SENT18 REVISION: His films include "Blue Story". |

Table 1: Sample datapoint with input-target formatting from the USB dataset

where the output probabilities fall below a threshold $\tau$ to select alternate decoding paths. Varying $\tau$ allows us to make more or fewer edits, effectively trading recall against precision. This approach produces a better precision-recall frontier than a baseline of randomly selecting additional words to edit to boost recall.

Our contributions are as follows:

- We present GENAUDIT, a tool to assist fact-checking LLM outputs in document-grounded tasks. The tool identifies and fixes errors, and highlights evidence for claims.

- We evaluate and release fine-tuned LLMs which serve as backend models for fact-checking; these perform comparably to SOTA proprietary LLMs in few-shot settings.

- We evaluate GENAUDIT for fact-checking errors in summaries generated by 8 LLMs for documents from 3 domains.

- We present and evaluate a custom decoding algorithm that allows one to improve error detection recall while incurring a smaller drop in precision than baselines.

## 2  Background

We fine-tune LLMs to perform evidence extraction and claim editing, and use them as backend for GENAUDIT. In this section we provide an overview of the training dataset and models we use to power the underlying evidence extraction and factual error correction tasks.

### 2.1  The USB dataset

The USB dataset (Krishna et al., 2023) is composed of Wikipedia articles, their summaries and (human) annotations on them. The summaries have two versions: (i) An initial version which may have content that is unsupported by the article or contradicted by it, and (ii) An edited version which annotators have created by making minimal edits to the initial version to remove errors. Additionally, each sentence in the edited summary is linked to a minimal set of article sentences that provide sufficient evidence for all facts that it contains.

We format the dataset in a sequence-to-sequence format to use it for fine-tuning LLMs (Table 1). The input to the model starts with the task instruction. It is followed by the reference document where each sentence is prefixed by a sentence ID (e.g. SENT1, SENT2...). It is then followed by the summary sentences upto the sentence to be fact-checked (called the *claim*). The sentences preceding the claim are included so that relevant context from it (e.g. coreferences) can be used for better understanding of the claim. Finally, the claim is appended to the input. The target output consists of the two parts. The first part contains a list of sentence ids from the document which provide evidence for the claim, and the second part consists of a revised version of the claim which removes its unsupported information and replaces incorrect facts.

We use a custom split of the USB dataset for training and evaluating our model. We shuffle and divide the entire dataset into train, validation and test splits of size 94%, 3% and 3% of the full dataset.

| Model | Error Identification | | | Evidence | | |
|---|---|---|---|---|---|---|
| | Recall | Precision | F1 | Recall | Precision | F1 |
| **Finetuned decoder-only LLMs** | | | | | | |
| Falcon-7B | 69.03 | 61.54 | 65.07 | 59.85 | 54.23 | 56.90 |
| Llama2-7B | 74.85 | 39.19 | 51.44 | 68.03 | 68.47 | 68.25 |
| Mistral-7B | 80.53 | 73.34 | 76.77 | 72.25 | 86.66 | 78.80 |
| **Fine-tuned encoder-decoder LLMs** | | | | | | |
| Flan-T5-XL | 73.01 | 87.07 | 79.42 | 78.90 | 85.69 | 82.16 |
| Flan-T5-XXL | **80.38** | 84.50 | **82.39** | **81.46** | 85.96 | **83.65** |
| Flan-UL2 | 76.47 | **87.44** | 81.59 | 80.56 | **86.42** | 83.39 |
| **Few-shot prompted proprietary LLMs** | | | | | | |
| GPT-3.5-turbo (8shot) | 38.79 | 48.57 | 43.13 | 51.79 | 45.15 | 48.24 |
| GPT-4 (4shot) | 37.98 | 63.89 | 47.64 | 74.42 | 38.52 | 50.76 |

Table 2: Performance of different models on the test split of the USB dataset for the tasks of (i) identifying erroneous words, and (ii) highlighting relevant evidence

This differs from the original USB splits in two ways. First, the training split is much larger at $94\%$ instead of the original $40\%$. Second, the training split consists of articles from all 6 domains in the benchmark, whereas originally 2 of the domains where reserved as challenging OOD examples to occur in only the test set. The motivation for both of these changes is that we want to create a tool which generalizes to other diverse data beyond simply Wikipedia articles, and hence we train on as much and as diverse data as possible.

## 2.2 Reducing memory requirement for training

We aim to fine-tune large models for the fact-checking task, since they are more likely to generalize better to unseen domains due to internal knowledge. Additionally, we need to feed in the entire reference document to the model, which can be thousands of tokens long. Both these factors increase the memory requirement for training the models, and to address this challenge, we use low-rank adapters with 4-bit quantization (Dettmers et al., 2023). Low rank adapters (Hu et al., 2021) reduce the memory requirement during training by reducing the number of trainable parameters, thus reducing gradient computation. To reduce the sequence length in cases where the reference document (Wikipedia article) is too long, we iteratively drop sections in it which are not relevant to any sentence in the summary (i.e. do not provide any evidence for it). We follow this process until the input sequence length is reduced to within a maximum limit, which is kept the same regardless of what model we are fine-tuning. We also use gradient accumulation and gradient checkpointing (Chen et al., 2016) to reduce memory footprint for training. For more details, please see the Appendix.

## 3 Experiments

We use the USB dataset for training/prompting and evaluating 8 different models for two factchecking tasks. For the evidence extraction task, we report precision, recall and F1 score as in a binary classification task where given a claim and reference document, each sentence in the reference is classified as relevant evidence or not. To evaluate the model's ability to remove errors, we compare the words removed/replaced by the model vs those removed in the ground truth revision. Given the original claim and a revision, we tokenize each into words and compute the diff between them (using Python's showdiff library). The words in the claim that are removed/replaced in the revision are tagged as incorrect and the remaining words are tagged as correct. We use the ground truth revision in the dataset to compute the ground truth tags, the model-generated revision to compute the predicted tags, and compute the corresponding precision, recall and F1 scores. Notably, it is difficult to compare the replacement text proposed by the model with ground truth replacements automatically, since the underlying text span being replaced must match exactly to make an aligned comparison. This requires human evaluation which we discuss in the next section.

We fine-tune and evaluate 6 different models. These include decoder-only models from the Falcon (Almazrouei et al., 2023), Llama2 (Touvron et al., 2023) and Mistral (Jiang et al., 2023) series, and encoder-decoder Flan-T5 models (Chung et al., 2022; Tay et al., 2022). Finally, we also use

| | Error Identification | | | | Replacements | Evidence Extraction | | | |
|---|---|---|---|---|---|---|---|---|---|
| | BaseRate | Recall | Precision | F1 | Accepted% | Recall | Precision | F1 | Sufficient% |
| Aggregate | 3.97 | 40.37 | 95.04 | 56.66 | 78.18 | 90.83 | 95.22 | 92.97 | 85.98 |
| **Summary generation models** | | | | | | | | | |
| Llama2-7B | 4.29 | 30.21 | 89.29 | 45.15 | 66.67 | 90.71 | 96.12 | 93.33 | 86.65 |
| Mistral-7B | 1.99 | 23.83 | 92.00 | 37.86 | 40.00 | 91.43 | 95.80 | 93.57 | 87.59 |
| Falcon-7B | 21.84 | 47.95 | 97.46 | 64.28 | 86.36 | 84.65 | 87.08 | 85.85 | 76.77 |
| Llama2-70B | 3.29 | 43.38 | 95.93 | 59.75 | 100.00 | 91.98 | 92.09 | 92.04 | 86.10 |
| Flan-UL2 | 9.68 | 34.04 | 96.04 | 50.26 | 80.00 | 90.18 | 93.96 | 92.03 | 84.16 |
| Gemini-pro | 1.80 | 27.75 | 78.69 | 41.03 | 66.67 | 91.27 | 96.98 | 94.04 | 86.68 |
| GPT-3.5-turbo | 1.10 | 29.06 | 89.47 | 43.87 | 75.00 | 92.32 | 97.23 | 94.71 | 88.09 |
| GPT4 | 2.53 | 56.77 | 100.00 | 72.42 | 87.50 | 90.39 | 97.07 | 93.61 | 85.49 |
| **Datasets** | | | | | | | | | |
| XSum | 4.54 | 55.00 | 98.69 | 70.64 | 77.78 | 94.81 | 97.92 | 96.34 | 92.83 |
| ACIBench | 2.73 | 44.04 | 90.65 | 59.28 | 88.89 | 87.57 | 93.30 | 90.34 | 80.84 |
| Reddit | 4.88 | 22.52 | 92.41 | 36.22 | 60.00 | 91.93 | 95.50 | 93.68 | 86.55 |

Table 3: Results from human evaluation of GENAUDIT predictions (using fine-tuned Flan-UL2 model). Aggregate results show performance on all evaluated document-summary pairs, whereas other rows show performance (a) on summaries generated using individual LLMs, or (b) on documents from specific datasets. BaseRate represents the percentage of tokens in summaries that are hallucinations. Sufficient% represents the percentage of summary sentences for which the evidence shown by the model was sufficient for deciding factuality.

OpenAI's GPT-3.5-turbo and GPT-4 models for the task via few-shot prompting, using 8 and 4 exemplars respectively.

We find that there is large variation in performance of decoder-only LLMs (Table 2). Llama2 outperforms Falcon in evidence extraction, but underperforms it in the claim editing task, due to its low precision. Mistral outperforms both Falcon and Llama2 models by a large margin. There is relatively less variation in performance of the three encoder-decoder models. Flan-T5-XXL and Flan-UL2 models perform the best, with the former providing better recall and the latter providing better precision on both tasks. Few-shot prompted GPT models perform worse than all fine-tuned models on both error identification and evidence extraction.

## 4 Human Evaluation

In the previous section we saw that models fine-tuned on the USB dataset perform well when evaluated on its test split. However, this does not imply that they would also perform well when deployed in diverse out-of-domain scenarios. Two types of domain shift can occur here. The first is a change in the domain of reference documents used for fact-checking. USB consisted of Wikipedia articles only, but we would ideally want a finetuned model to work with other document types such as news articles or meeting transcripts. Another sort of domain shift is the specific model generating the content to be fact-checked. USB consists only of claims written by humans, but we would want models to detect and fix errors in content generated by a arbitrary LLMs.

We run experiments to evaluate the performance of GENAUDIT when fact-checking summaries generated by different models for documents sampled from different domains. We include a diverse set of open-source models, including three decoder-only 7B parameter LLMs (Mistral (Jiang et al., 2023), Llama2 (Touvron et al., 2023), Falcon (Almazrouei et al., 2023)), one large 70B parameter model (Llama2), and one encoder-decoder model (Flan-UL2 (Tay et al., 2022)). As proprietary API-based models, we use GPT-3.5-turbo, GPT-4, and Gemini-pro models for summary generation. We then use the Flan-UL2 model finetuned on USB to fact-check the generated summaries.

We select documents for summary generation from the following three datasets.

**XSum** (Narayan et al., 2018)    A summarization dataset consisting of BBC news articles covering diverse topics and events.

**ACI-Bench** (Yim et al., 2023)    A dataset for summarization of patient visits to the doctor comprising transcripts of doctor-patient encounters.

**Reddit-TIFU** (Kim et al., 2019)   A dataset consisting of posts from the online discussion forum Reddit in which users narrate personal day-to-day experiences.

We randomly select 30 documents from each of the three datasets for which to generate summaries. For the Reddit-TIFU dataset, we manually filtered out examples containing profanity or sexually explicit content. While generating summaries with open-source models, we decode using top-$p$ nucleus sampling (Holtzman et al., 2019) from the output token distribution with a top-$p$ value of $0.9$ and a temperature of $1.0$.

We hired annotators via Upwork,[3] and instructed them to evaluate all edits suggested by the fact-checker, accept those that fix legitimate factual errors, and mark incorrect suggestions. Annotators were also instructed to find any missing errors in summaries which were not highlighted by the system, and to fix them by making minimal edits.

To provide feedback on the highlighted evidence, annotators provided binary (relevant/not relevant) feedback for each suggested evidence sentence in the summary. They were also instructed to consider if the highlighted evidence for each summary sentence is sufficient or not; of not, then they should mark additional source sentences which contain the missing evidence. Additionally, we asked annotators to flag incomprehensible summaries, which were then excluded from the analysis. For example, instead of a summary, sometimes Falcon-7B model outputs a continuation of instructions, such as *"when creating a summary, use the information given and avoid superfluous details."* The Appendix includes additional evaluation details.

Results are shown in Table 3. We use the metrics described in Section 3 for rating suggested errors and evidence generated by GENAUDIT. On an aggregate level—across all domains and summary generation models—GENAUDIT identifies erroneous words with high precision ($95.04\%$) and moderate recall ($40.37\%$). Note that achieving high recall is challenging here given the low prevalence of erroneous words ($3.97\%$). With respect to evidence extraction, we observe high precision and recall ($95.22\%$ and $90.83\%$, respectively), suggesting that most evidence sentences highlighted by the model are useful for fact-checking the given claim, and only few evidence sentences are missed (not highlighted) by the model.

The rate of errors in outputs varies considerably across models. Summaries from GPT-3.5-turbo have the lower error rate at $1.10\%$, while Falcon-7B has the highest error rate of $21.9\%$. The highest recall and precision for error detection is observed for the latter model. The precision of error detection remains around or above $90\%$ for all models except Gemini-pro. Recall varies widely ($\sim23 - 57\%$) across different models. The lowest recall is for Mistral-7B ($23.83\%$); for context, its error rate is $1.99\%$. For evidence extraction, the performance with most models is quite similar with both precision and recall, falling between $\sim85 - 97\%$. The lowest F1 score is $85.85\%$ for Falcon-7B, and highest is $94.71\%$ for GPT-3.5-turbo.

Among the datasets considered, GENAUDIT's performance at error identification was best for XSum (news articles), followed by ACIBench (clinical conversations), and finally Reddit/TiFU (social media posts). While the precision stays above $90\%$ on all datasets, the recall ranges from $22.52\%$ on the Reddit dataset to $55.00\%$ on XSum. On the evidence extraction task, GENAUDIT achieves F1 scores of $90\%+$ on all three datasets.

While the previously discussed metrics measure success at identifying parts of the text which are incorrect, we also measure the quality of model-generated replacements when they are suggested. The percent of model-suggested replacements accepted was $\sim78\%$, on average (Table 3), suggesting the quality of generated replacement strings. The percent of generated summary sentences for which the highlighted evidence was sufficient for verification was $\sim86\%$, indicating that generated evidence highlights may make fact-checking more efficient.

Using the annotations collected above, we evaluate additional models on the error detection task. Few-shot prompted GPT-4 achieves better recall than Flan-UL2, while other fine-tuned models achieve lower recall (Table 4). All models achieve lower precision than Flan-UL2. Edits suggested by GPT-4 add about $8.8\%$ more words to them on average, while other fine-tuned models add a negligible amount, reflecting the tendency of GPT-4 to make substantial changes to text. Finally, we also evaluate the FAVA model trained by Mishra et al. (2024) for factual error correction. We see

---

[3] `https://www.upwork.com/`

| Model | Recall | Precision | %Del | %Add |
|-------|--------|-----------|------|------|
| Flan-UL2 | 40.37 | **95.04** | 1.69 | 0.18 |
| Flan-T5-XL | 25.75 | 74.23 | 1.38 | 0.12 |
| Mistral-7B | 35.08 | 45.24 | 3.08 | 0.16 |
| GPT-4 (4-shot) | **40.68** | 28.50 | 5.67 | 8.80 |
| FAVA (Mishra et al., 2024) | 14.18 | 31.34 | 1.80 | 0.43 |

Table 4: Performance of models fine-tuned by us on the USB dataset, few-shot prompted GPT-4, and the FAVA model (Mishra et al., 2024) at identifying erroneous words in model-generated summaries, along with the percentage of summary words deleted and added by their edits.

---

**Algorithm 1** Thresholded Edit

**Input:** document $D$, claim $C$, predicted evidence $E$, predicted revision $R$, model $\mathcal{M}$, threshold $\tau$
$Q = (D, C, E)$
$t = 1$
**while** $t \leq |R|$ **do**
     // run this loop until the timestep counter reaches end of string R
     $p_1 p_2 ... p_{|V|} = \textbf{NextTokProb}_{\mathcal{M}}(r_1 .. r_{t-1} \mid Q)$      // run M to get probability of tokens that can be generated at timestep t
     **if** $p_{r_t} \leq \tau$ **then**
         // if the probability of the token currently at timestep t in R is lower than tau, we replace it
         $r' = \arg\max_k(p_k \mid k \neq r_t)$      // replace it with the token with the next highest probability (call it r')
         $\text{prefix} = r_1 r_2 .. r_{t-1} r'$
         $\text{compl} = \textbf{Generate}_{\mathcal{M}}(\text{prefix} \mid Q)$      // use M to generate the full completion of the revision if we use r' at timestep t
         $R' = \text{prefix} + \text{compl}$      // we call the new revision R'
         $N_{del}, N_{add}, \text{repl} = \textbf{DiffAtPos}_t(R, R')$      // extract span-level change between R and R' at timestep t (deletions and replacements)
         $R = \text{prefix} + \text{repl} + r_{(t+N_{del})} ... r_{|R|}$      // commit that change at t in R and discard any other changes observed later
         $t = t + N_{add}$
     **end if**
     $t = t + 1$      // advance the pointer and repeat the loop to find more spans to edit in R if they exist
**end while**
**Output:** updated revision $R$

---

that the model achieves the lowest recall compared to all the models evaluated, with a slightly higher precision than GPT-4.

## 5 Improving Recall of Error Detection

Users fact-checking LLM outputs using GENAUDIT may give more importance to a higher recall than precision to be confident that most errors are highlighted for review, even at the cost of false positives. While it is always possible to increase recall by indiscriminately flagging additional text spans as errors, a naive strategy would lead to a large drop in precision. We propose a decoding algorithm for the fact-checking model which uses the output token probabilities to achieve a better precision-recall trade-off.

Our proposed approach (Algorithm 1) for increasing error detection recall relies on observing the probabilities of tokens generated as the revision by the fact-checker, and intervening at timesteps with low model confidence. Given a document $D$, claim $C$, and an initially generated revision $R = r_1 r_2 .. r_m$, we find the first position $t$ where the probability assigned to token $r_t$ falls below a threshold $\tau$. At that timestep we then generate the token with the highest probability *excluding $r_t$*. We generate the remaining tokens (from $t + 1$) as usual via greedy decoding to compute an alternate revision $R'$. Given $R$ and $R'$, assume the span $r_k r_{k+1} .. r_{k+w}$ was replaced by $x_1 .. x_q$. We make the replacement in $R$ yielding $r_1 .. r_{k-1} x_1 .. x_q r_{k+w+1} ... r_m$. We then repeat the process of finding low probability tokens and making edits for the remaining tokens. After each iteration of the while loop, the value of $(|R| - t)$ decreases by at least 1, which guarantees termination of the program.

Increasing the value of $\tau$ in Algorithm 1 would lead to more edits being made to the claim, and vice-versa. We run the Flan-UL2 fact-checking model with different values of $\tau$ ranging from 0.0 to

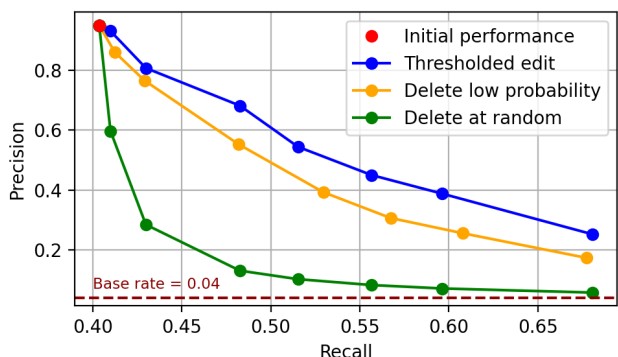

Figure 2: Variation in precision and recall of error identification by a fine-tuned Flan-UL2 model when using thresholded editing (Algorithm 1), versus editing out additional tokens either at random or by selecting the ones with low probability.

0.99 and plot the resulting recall and precision at detecting errors annotated in the human evaluation experiment in Section 4. We find that using Algorithm 1, we are able to increase the recall from about 40% to 60%, with a drop in precision from 95% to 40%. Although there is a drop in precision, the drop is much lower than what one would get by using a simple randomized baseline. The baseline strategy we compare against is to boost recall by randomly selecting additional words (beyond the ones already predicted as erroneous by the model) and mark them as erroneous too. We compute the number of words that need to be selected to boost expected recall to a certain level, and the resulting drop in expected precision that it entails (see Appendix for derivation). The thresholding approach maintains a much higher precision with increasing recall compared to the baseline strategy, where the precision already falls to around 28% when the recall increases to merely 43% (Figure 2).

We also compare using the custom decoding strategy in Algorithm 1 with simply flagging additional tokens as non-factual if their probability of generation (by the fact-checking model) falls below a variable threshold. We see that this strategy performs worse than using Algorithm 1 (Figure 2). This suggests that post-hoc usage of token probabilities from the fact-checking model does not isolate the non-factual spans as well as active intervention during the decoding process as done in Algorithm 1.

## 6 Related Work

Multiple works have focused on the task of classifying whether a generated summary contains any factual error or not, using trained models (Kryściński et al., 2020; Goyal and Durrett, 2021), question-answering based approaches (Fabbri et al., 2022), or prompting LLMs (Laban et al., 2023). While these works predicted factual correctness with respect to a given source document, recent works have implemented fact-checking against a large corpus (Wikipedia) by combining evidence retrieval and factuality prediction (Kamoi et al., 2023; Min et al., 2023). Our effort goes beyond binary prediction of factual correctness, by also localizing the errors in the claims and fixing them via minimal editing. A concurrent work from Mishra et al. (2024) also attempts to fix factual errors via editing, and we compared the performance of their released model with ours in Section 4. Liu et al. (2023) and Krishna et al. (2023) introduced the DeFacto and USB datasets respectively with human annotations to train and evaluate models for revising incorrect claims, and extracting evidence from a reference document. While both datasets can potentially be used to train GENAUDIT backend models, we used the USB dataset because (1) it contains comprehensive evidence labels for *all* facts in the claim, and (2) it contains multi-sentence summaries, which are more common in practice. We extend these lines of work by contributing an interactive tool for fact-checking, and a comprehensive evaluation of the models trained on such data at fixing errors in modern LLM outputs with evidence.

## 7 Conclusion

We introduced GENAUDIT, a tool to assist users in fact-checking LLM generated outputs against inputs by presenting supporting evidence and highlighting (and fixing) errors. We trained models for

fact-checking tasks which rival few-shot prompting of SOTA LLMs, and designed a web-interface for users to interact with. We evaluated GENAUDIT for fact-checking summaries generated by 8 LLMs for documents in 3 domains. Finally, we proposed a decoding algorithm for our fact-checking model to improve the recall of error identification while minimizing the cost in precision.

## Acknowledgements

This work was supported in part by the National Science Foundation (NSF), grant RI: 2211954 and 2211955.

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

# A Appendix

## A.1 Binary Classification of Factuality

In the main body of the paper, we evaluated the performance of GENAUDIT at localizing factual errors within text and suggesting edits. However, it can also be repurposed as a binary classifier which simply predicts whether a long-form generated text is factually consistent or not with respect to given reference. To do that, we simply declare a given passage of text as factually inconsistent with respect to reference document if GENAUDIT suggests any edit to any sentence in it.

We evaluate the performance of GENAUDIT on the SummEdits benchmark (Laban et al., 2023) which consists of document-summary pairs where the summaries potentially contain factual errors. The source documents are taken from 10 different datasets representing a diverse group including legal documents, scientific papers, emails etc. We tokenize the source document into individual sentences before passing it through the fact-checking model. GENAUDIT achieves a balanced accuracy score of 74.7, outperforming many LLMs and traditional fact-checking methods, with the exception of Gemini-pro and GPT-4 (Table 5)[4].

| Model | Balanced Accuracy |
|---|---|
| Human Performance | 90.92 |
| GPT4 | 82.06 |
| Gemini-pro | 75.49 |
| **GENAUDIT** | **74.75** |
| Claudev21 | 74.36 |
| Claudev2 | 73.58 |
| ChatGPT | 71.18 |
| PaLM-bison | 69.04 |
| QAFactEval (Fabbri et al., 2022) | 65.46 |
| Llama2-13b | 58.35 |
| Mistral-7b | 57.78 |
| SummaCConv (Laban et al., 2022) | 57.14 |
| DAE (Goyal and Durrett, 2021) | 55.17 |
| Llama2-7b | 50.36 |

Table 5: Performance of models on the SummEdits benchmark for binary classification of factuality. Here GENAUDIT uses the fine-tuned Flan-UL2 backend, whereas other LLMs are zero-shot prompted.

## A.2 Details on human evaluation

We engaged annotators via Upwork[5], leveraging their expertise to evaluate model-generated summaries against source documents. Candidates were chosen through a qualifying round, focusing on their ability to identify inaccuracies in summaries. Ultimately, two proficient proofreaders and fact-checkers were selected based on their performance on the qualifying task. Each annotator was

---

[4]Values taken from the official Github repository `https://github.com/salesforce/factualNLG`

[5]`https://www.upwork.com/`

tasked with annotating all summaries for half of the documents in each of the 3 datasets used (see Section 4). Annotators received compensation at an average rate of $25 USD per hour for their contributions.

We use a slightly modified version of the GENAUDIT UI to collect the annotations, with two notable changes (Figure 3). First, we add a buttons next to every source sentence to provide a accept/reject feedback if the source sentence is highlighted as evidence for a summary sentence. For source sentences which are not highlighted as evidence, we provide an accept button to mark it as additional evidence if needed. Second, the UI enables cycling through multiple model-generated summaries for the same source document at once. This is done to save annotators' time, since otherwise the annotators would have to read the source document again each time they get a summary for it in the sequence of annotation jobs. The annotators were instructed to mark the following categories of generated summary sentences as *invalid* which we excluded from our analysis: (i) truncated sentences, which occur due to the maximum decode length limit being reached. (ii) incomprehensible sentences, such as when the models generate instructions instead of summary (e.g. Falcon-7B once generated *"when creating a summary, use the information given and avoid superfluous details."*)

## A.3 Annotator Instructions

The following protocol was provided to annotators for assessing both the supporting evidence for summary sentences, and their factual consistency:

*Evidence Annotation*:

*a) Evaluate each summary sentence by reviewing all linked evidence from the source, marking them as either accepted (by clicking a tick) or rejected (by clicking a cross) based on their validity.*

*b) Examine the summary for unsupported information with respect to suggested evidence, If any information supporting the summary is found in the source but is not already supported by the suggested evidence, mark it as "new" evidence. Note that "new" evidence pertains only to instances where the summary includes seemingly unsupported information compared to the suggested evidence.*

*Summary Annotation*:

*a) First, accept all suggestions made by the tool that are **valid**. This encompasses deletions made by the tool of unsupported information and any recommended edits or replacements.*

*b) If there is incorrect or contradictory information in the summary not satisfactorily addressed by the tool then make minimal edits to the summary to align it with the source. If the edit is based on source information not already included in suggested evidence, label it as "new evidence."*

*c) For cases where the summary introduces unsupported information not addressed by the tool, remove corresponding segments of the summary without altering the evidence.*

*Note if a summary sentence is incomplete or incomprehensible, mark the sentence as INVALID.*

## A.4 Inter-annotator agreement

For one-third of the documents in each dataset, we got their model-generated summaries annotated by both annotators to estimate inter-annotator agreement. The doubly-annotated data included 232 summaries consisting of a total of 989 sentences. We compute the agreement on error identification by comparing the words removed/replaced from the initial summary by the two annotators. The value of Cohen's Kappa for this is 62.7 indicating substantial agreement. Similarly, we compare the ratings (useful vs not) provided to each suggested evidence sentence by the two annotators, which yields a Cohen's Kappa value of 58.56 indicating moderate agreement.

## A.5 Derivation of baseline precision-recall trade-off

Assume a binary classification problem with a dataset of $T$ datapoints, out of which $P$ have positive labels and $N$ have negative labels. Let's assume a prediction model assigns positive labels to $P'$ datapoints, and achieves a recall of $\alpha$ and precision of $\beta$. We want to boost the recall to a target of $\alpha'$ by flipping the predictions from negative to positive for some datapoints. In this section we derive the

precision-recall trade-off achieved if we select the labels to flip uniformly at random, which is the baseline used in Section 5.

If we flip the labels of $k$ datapoints from negative to positive, the expected number of them which would be true positives will be

$$\Delta = \frac{P(1-\alpha)}{(T-P')}k$$

For an expected target recall of $\alpha'$, we get

$$\alpha' = \frac{\alpha P + \Delta}{P}$$

$$\implies \alpha' = \alpha + \frac{k(1-\alpha)}{(T-P')}$$

$$\implies k = \frac{\alpha' - \alpha}{1-\alpha}(T-P')$$

The expected true positives is $\alpha'P$, which yields the new expected precision $\beta'$ to be

$$\beta' = \frac{\alpha'P}{(P'+k)}$$

$$\implies \beta' = \frac{\alpha'P}{P' + \frac{(\alpha'-\alpha)}{(1-\alpha)}(T-P')}$$

Note that $P = \gamma T$, where $\gamma$ is the base rate for positive class, and the number of predicted positive labels $P' = \frac{\alpha P}{\beta}$. Substituting these, we get our final form

$$\beta' = \frac{\alpha'\gamma}{\frac{\alpha\gamma}{\beta} + \frac{(\alpha'-\alpha)}{(1-\alpha)}\left(1 - \frac{\alpha\gamma}{\beta}\right)}$$

We showed the performance of this baseline against our proposed Algorithm 1 in Figure 2.

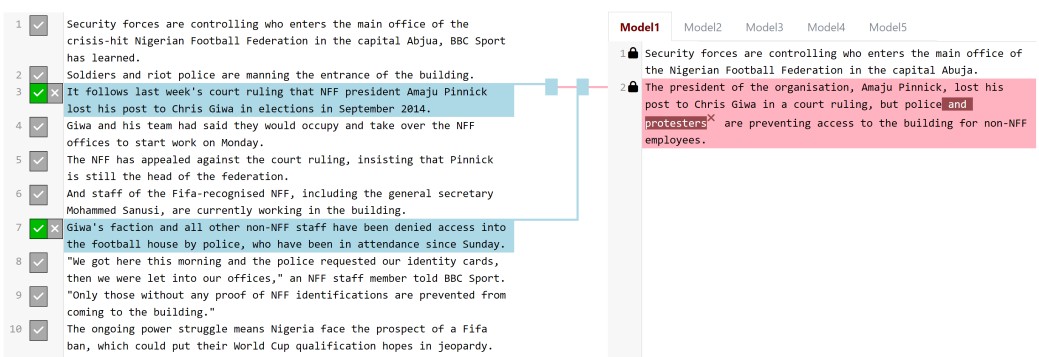

Figure 3: Interface used for collecting feedback on suggested evidence and edits from GENAUDIT. Annotators can accept/reject each suggested evidence sentence, and can also mark additional sentences as evidence if needed. Suggested edits can be accepted/rejected by clicking on the button on the top-right of the highlighted span. if needed, users can also make freeform edits to fix more errors. Annotators cycle through the summaries generated by different models, whose names are anonymized and their order is shuffled.

| Model | License | URL |
|---|---|---|
| Llama2-7B | llama2 | https://huggingface.co/meta-llama/Llama-2-7b-chat-hf |
| Llama2-70B | llama2 | https://huggingface.co/meta-llama/Llama-2-70b-chat-hf |
| Mistral-7B | Apache-2.0 | https://huggingface.co/mistralai/Mistral-7B-Instruct-v0.1 |
| Falcon-7B | Apache-2.0 | https://huggingface.co/tiiuae/falcon-7b-instruct |
| Flan-T5-XL | Apache-2.0 | https://huggingface.co/google/flan-t5-xl |
| Flan-T5-XXL | Apache-2.0 | https://huggingface.co/google/flan-t5-xxl |
| Flan-UL2 | Apache-2.0 | https://huggingface.co/google/flan-ul2 |
| GPT-3.5-turbo | openai | https://platform.openai.com/docs/models/gpt-3-5-turbo (version gpt-3.5-turbo-16k-0613) |
| GPT-4 | openai | https://platform.openai.com/docs/models/gpt-4-and-gpt-4-turbo (gpt-4-0613) |

Table 6: Links to models used in Section 3 and Section 4

| Dataset | License | URL |
|---|---|---|
| USB | Apache-2.0 | https://huggingface.co/datasets/kundank/usb |
| XSum | Unknown | https://huggingface.co/datasets/EdinburghNLP/xsum |
| ACIBench | CC-BY-4.0 | https://github.com/wyim/aci-bench |
| Reddit-TIFU | MIT | https://huggingface.co/datasets/reddit_tifu |
| SummEdits | Apache-2.0 | https://github.com/salesforce/factualNLG |

Table 7: Links to datasets used in this work

## A.6 Training details

Each model fine-tuned on the USB dataset, was trained for 10 epochs and the best checkpoint was selected based on validation performance at the end of each epoch. The effective batch size was kept at 128. The optimizer used was 8-bit AdamW with $\beta_1 = 0.9$ and $\beta_2 = 0.999$, without weight decay, and with a learning rate of $5e-5$. The maximum input and output sequence lengths were set to 3050 and 150 tokens respectively. We used 4-bit quantization with low-rank adapters (Dettmers et al., 2023) with parameters $r = 8$ and $\alpha = 32$ .

Each training run was carried out on $2\times$ Nvidia A6000 GPUs each of which has 49GB of memory. We used gradient accumulation and gradient checkpointing to reduce the peak memory requirements during training. The duration of each training run varied depending on the model used but fell in the range of 30-70 hours. The implementation was done using the Huggingface transformers (Wolf et al., 2019), Deepspeed (Rasley et al., 2020), Peft (Mangrulkar et al., 2022) and Bitsandbytes[6] libraries.

## A.7 Sample model outputs

We show some sample corrections suggested by GENAUDIT using the Flan-UL2 backend to summaries generated by LLMs in Figures 4, 5, 6, 7.

## A.8 Limitations

In this section, we highlight some limitations of our current work. Deciding whether a span of generated text is a hallucination involves subjective judgement. While some cases clearly fall in one category (e.g. if the model invents the name of a person), others can be quite debatable. For example, while summarizing a Reddit post about a person's bad experience, the summary mentioned them feeling *left out*, which was changed to *disappointed* by one annotator but was left as-is by the other annotator. Such decisions are motivated by the annotators' subjective judgement of whether the inferences made in the summary are consistent with the source, leading to difference in the edits made by them. This leads to the lack of a single gold ground-truth, which makes automatic evaluation challenging.

We used the edits made to LLM-generated summaries by human annotators as gold reference while evaluating different models' performance at error identification (Table 4). During the process, the annotators were shown suggested evidence and edits from the best-performing fine-tuned model (Flan-UL2) to evaluate its quality and to assist in the annotation process. Using the predictions from a model to assist collecting ground truth labels may lead to a benchmark that favors it and similar models. Ideally, we should conduct separate human evaluation for each model's predictions where annotators look at the suggestions from that model before making edits. However, that would be

---

[6]https://github.com/TimDettmers/bitsandbytes

| Model | Prompt |
|-------|--------|
| GPT-4 | You are provided a document and its summary. The summary may potentially contain facts which contradict with the document or are not supported by any evidence in the document. The last sentence of the summary is marked as a claim. Find and list sufficient sentences in the document to provide evidence for the claim. Make sure to provide evidence for all the supported facts in the claim. Then, revise the claim to remove or replace facts which are not supported by the document or are contradicted by it. Only make changes to the text of the claim when necessary. When you add new information to the claim, it must be only to fix a contradictory fact in the claim, and not for changing the style of the text. |
| GPT-3.5-turbo | You are provided a document and its summary. The summary may potentially contain facts which contradict with the document or are not supported by any evidence in the document. The last sentence of the summary is marked as a claim. Find and list sufficient sentences in the document to provide evidence for the claim, and then revise the claim to remove or replace facts which are not supported by the document or are contradicted by it. When you add new information to the claim, it must be only to fix a contradictory fact in the claim, and not for changing the style of the text. |
| Others | You are provided a document and its summary. The summary may potentially contain factual errors. The last sentence of the summary is marked as a claim. Find all sentences in the document providing evidence for the claim, and then revise the claim to remove or replace unsupported facts. |

Table 8: Prompts used for fact-checking using GPT models in zero-shot setting, and using other models with fine-tuning (as described in Section 3)

| Model | Prompt |
|-------|--------|
| Gemini-pro, GPT-3.5-turbo, GPT-4 | Generate a summary for the following document in brief. When creating the summary, only use information that is present in the document. Generate the summary in free-form text without using bullet points. |
| Others | Generate a summary for the following document in brief. When creating the summary, only use information that is present in the document. |

Table 9: Prompts used for summary generation using different models for human evaluation experiment (Section 4)

prohibitively expensive and time-consuming, and so we conduct automatic evaluation instead by re-using the collected annotations as ground truth.

### A.9 Ethical Considerations

To evaluate GENAUDIT, we recruited proficient proofreaders who were selected after a qualifying round, focusing on their ability to identify inaccuracies in summaries. Annotators received compensation at an average rate of $25 USD per hour for their contributions. Annotators were provided the opportunity to discuss their concerns and questions with the authors throughout the annotation process.

The models presented in this work for fixing factual errors in LLM outputs are not perfect and some errors may not be detected by it. Hence, in critical application areas (such as clinical settings), it should be used in conjunction with a human verifier who uses GENAUDIT as a tool rather than as a perfect error detector. We have made this clear by transparently providing the performance of GENAUDIT via human evaluation in Section 4.

We verified that the different models and datasets in this work have licenses that permit research use.

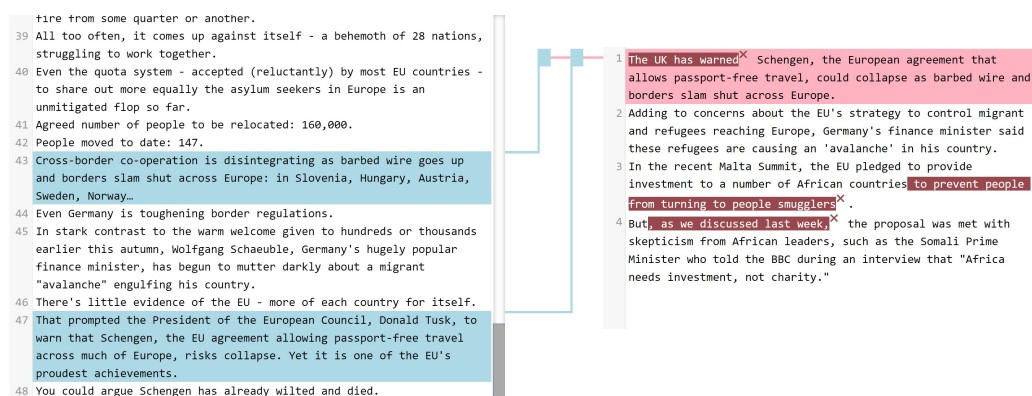

Figure 4: GENAUDIT suggested edits for a GPT4-generated summary of a news article from the XSum dataset. The first sentence contains a statement attributed to the UK which was actually made by the president of the European Union

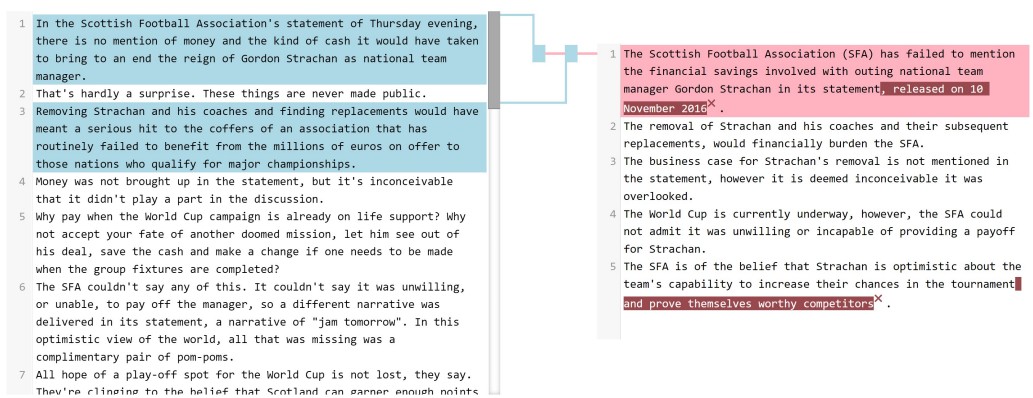

Figure 5: GENAUDIT suggested edits for a GPT4-generated summary of a news article from the XSum dataset. Interestingly, The reference document does not contain the date on which the said statement was made by SFA. Interestingly though, GPT-4 almost got it right by sheer memorization. The statement was released on 17 November 2016 whereas GPT-4 mentioned 10 November 2016. (Ref: `https://www.bbc.com/sport/football/38021627,https://www.bbc.com/sport/football/38019477`)

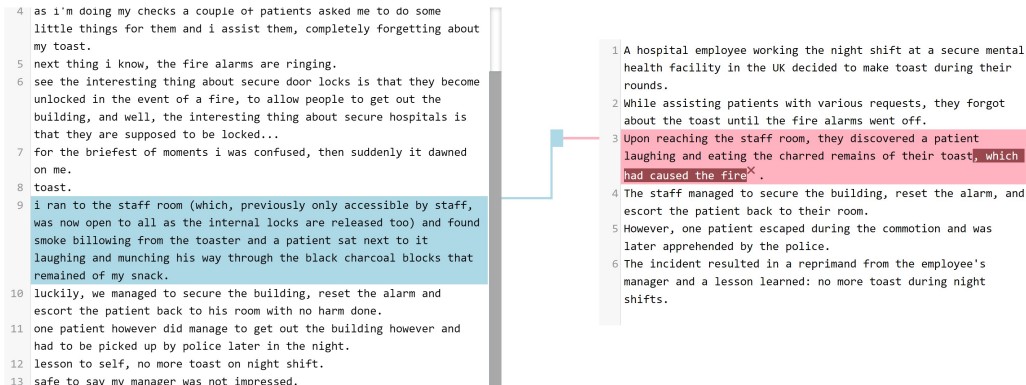

Figure 6: GENAUDIT suggested edits for a Geminipro-generated summary of a Reddit post where a person describes fire alarm going off at a workplace due to smoke from burnt toast. The summary suggests that there was a fire caused which doesn't seem to be the case from the reference.

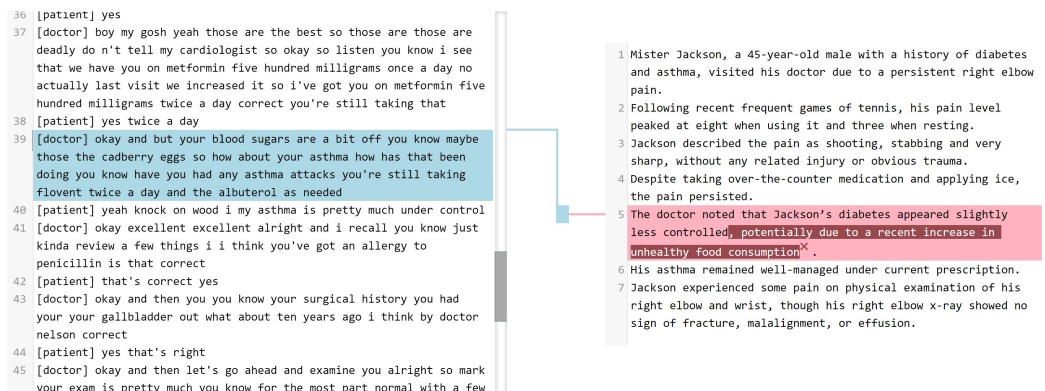

Figure 7: GENAUDIT suggested edits for a GPT4-generated summary of a conversation between a doctor and patient. Here, the doctor briefly mentions that the patient's blood sugar problems may be caused by eating chocolates, but they don't suggest that such unhealthy consumption has increased recently (as the summary claims).

