# OpenReview forum: "GenAudit: Fixing Factual Errors in Language Model Outputs with Evidence"
_NeurIPS.cc/2024/Workshop/SafeGenAi — SafeGenAi Poster_

### Official Review · Reviewer_vaaH · 2024-10-09

**Rating:** 7
**Confidence:** 3

**Review:**

**Short Summary: **This paper introduces the tool GenAudit for fact-checking and modifying LLM responses.

**Quality and Clarity:** I found no problems with the clarity or quality of the paper.

**Originality:** It is novel to introduce a tool to suggest edits based on the fact-checking LLM and allow these fact-checking LLMs to be evaluated. Additionally, they compared against a similar hallucinating detection approach (Mishra et al., 2024).

**Strengths:**
- This paper presents an important tool for using and evaluating fact-checking LLM, which is relevant for safe generative AI.
- The experiments are well-documented and sound.
- The human evaluation sufficiently satisfied my concerns about changing the train, validation, and test USB dataset splits and the error identification rate.
- Recall vs. precision is an important topic that is well addressed with the Thresholded Edit algorithm and analysis included in the main paper.

**Weaknesses:**
- While I understand what Algorithm 1 is doing, the middle portion of the algorithm was difficult to follow, even with the comments.
- It is also understandable to drop the irrelevant portions of the reference document when it is too long, but it would be interesting to know if the authors had any thoughts on how this would affect the results.

**General Review:** This paper provides a tool and evaluates LLMs for the important task of fact-checking, which can be used to verify other LLMs. It is well-written and sound, and the weaknesses above do not degrade its quality.

---

### Official Review · Reviewer_Nro1 · 2024-10-11
**Need to highlight the novelty**

**Rating:** 6
**Confidence:** 4

**Review:**

In this paper, the authors present a method for finding and fixing factual errors in LLM-generated content. I think the work is solid since they provide an end-to-end solution to this issue and even develop a system with a user interface. The human annotation part is also very meaningful for future research.

I have the following concerns.

We know that the probability of successful evidence-searching and fact-checking is greatly affected by the length of the reference document. The authors should provide more analysis/experimental results to validate the scalability of the proposed method.

In addition, there exist several different strategies to reduce factual errors. For example, we can leverage advanced  Retrieval Augmented Generation methods, which do not need fine-tuning and can also help us find evidence and correct the generation. The authors should explain more about this to highlight the unique advantages of the proposed method. Also, the authors just fine-tuned the LLMs. The novelty is limited.